# EPS Production by *Lacticaseibacillus casei* Using Glycerol, Glucose, and Molasses as Carbon Sources

**DOI:** 10.3390/microorganisms12061159

**Published:** 2024-06-06

**Authors:** Guilherme Deomedesse Minari, Rodolfo Debone Piazza, Daiane Cristina Sass, Jonas Contiero

**Affiliations:** 1Institute of Biosciences, São Paulo State University (Unesp), Rio Claro 13506-900, Brazil; gdminari@gmail.com (G.D.M.); daiane.sass@unesp.br (D.C.S.); 2Institute of Chemistry, São Paulo State University (Unesp), Araraquara 14800-903, Brazil; rodolfo.piazza@unesp.br; 3Institute on Research in Bioenergy, São Paulo State University (Unesp), Rio Claro 13500-230, Brazil

**Keywords:** extracellular polymer, glucose, mannose, galactose, arabinose, exopolysaccharides, *Lactobacillus*

## Abstract

This study demonstrates that *Lactobacillus* can produce exopolysaccharides (EPSs) using alternative carbon sources, such as sugarcane molasses and glycerol. After screening 22 strains of Lactobacillus to determine which achieved the highest production of EPS based on dry weight at 37 °C, the strain Ke8 (*L. casei*) was selected for new experiments. The EPS obtained using glycerol and glucose as carbon sources was classified as a heteropolysaccharide composed of glucose and mannose, containing 1730 g.mol^−1^, consisting of 39.4% carbohydrates and 18% proteins. The EPS obtained using molasses as the carbon source was characterized as a heteropolysaccharide composed of glucose, galactose, and arabinose, containing 1182 g.mol^−1^, consisting of 52.9% carbohydrates and 11.69% proteins. This molecule was characterized using Size Exclusion Chromatography (HPLC), Gas chromatography–mass spectrometry (GC-MS), Fourier-transform infrared spectroscopy (FTIR), and proton nuclear magnetic resonance spectroscopy (^1^H-NMR). The existence of polysaccharides was confirmed via FT-IR and NMR analyses. The results obtained suggest that *Lacticaseibacillus casei* can grow in media that use alternative carbon sources such as glycerol and molasses. These agro-industry residues are inexpensive, and their use contributes to sustainability. The lack of studies regarding the use of *Lacticaseibacillus casei* for the production of EPS using renewable carbon sources from agroindustry should be noted.

## 1. Introduction

Lactic-acid-producing bacteria (LAB) are microorganisms considered to be probiotics that can be used to obtain chemical metabolites, such as lactic acid, and generally considered safe. These bacteria are known to produce exopolysaccharides (EPSs), with 30 species of EPS synthesizers identified to date, the most well-known of which are *L. casei*, *Lactobacillus acidophilus*, *Lactobacillus brevis*, *Lactobacillus curvatus*, and *Lactobacillus delbrurckii* [1,2,3]. EPSs are polymers that occur naturally and are biodegradable and nontoxic. These polymers play crucial roles in various biological mechanisms, such as energy storage and the protection of bacterial cells from harsh environmental conditions, including exposure to antibiotics and digestive enzymes, such as bile salts and pancreatic enzymes. EPS molecules can tolerate high salt concentrations and a range of pH values and can even modulate the immune system of their host [1,2,4,5,6,7,8,9]. Recognized as prebiotics, these molecules can be used in the food, pharmaceutical, and cosmetic industries. In the food industry, EPSs serve as rheological agents that can alter the texture of fermented products by controlling their viscosity or stabilizing or emulsifying them or by serving as gelling agents [10,11,12,13]. In the pharmaceutical industry, EPSs are commonly associated with beneficial effects on human health and disease prevention due to their immunostimulatory, immunomodulatory, antitumor, antibiofilm, antioxidant, and cholesterol-lowering activities [11,13,14]. The synthesis of extracellular polymers is affected by several factors, such as the composition of the medium (carbon and nitrogen sources), pH, temperature, aeration, and the specific genes present in each species [15,16]. The carbon/nitrogen (C/N) ratio has a direct impact on the production of EPSs. The quantity of these substrates in a medium determines their rate of utilization, which, in turn, determines the rate of EPS production [15,17,18]. EPS production can be supported by glucose, fructose, sucrose, or lactose [2,13,14,19,20,21]. Obtaining biopolymers through the use of low-cost carbon sources is an alternative approach. EPS is made up of various biochemical groups that are secreted by different microorganisms, including prokaryotes and eukaryotes. This polymer mainly consists of polysaccharides or carbohydrates, proteins, lipids, phosphates, acetates, succinates, nucleic acids, and humic acids [22]. Carbohydrates and proteins constitute around 90% and 75% of an EPS molecule, respectively [17]. The chemical composition of an EPS determines its classification as a homopolysaccharide or heteropolysaccharide [23,24]. Homopolysaccharides are usually neutral and composed of L-fructose or D-glucose [3,17,25]. Polysaccharides are carbohydrates that can be found in different forms, such as dextran, cellulose, or curdlan, the last of which is a gum made up of (1,3)-β-d-glucan. Heteropolysaccharides, such as alginate, xanthan gum, gellan gum, benzol gum, cholanic acid, and hyaluronic acid, are usually produced by lactic-acid-producing bacteria. These types of polysaccharides are formed by repeating monosaccharide units (D-glucose, D-galactose, L-fructose, L-rhamnose, D-glucuronic acid, L-guluronic acid, and D-mannuronic acid) [17,26]. Several microorganisms produce polymeric compounds during metabolic processes that can be applied in different fields [21,27,28]. Species of *Lactobacillus*, specifically *Lacticaseibacillus casei*, have been found to produce biopolymers [2]. However, there is little research available on the production of EPS by *Lacticaseibacillus casei*. The most recent studies suggest that the biopolymer synthesized by *Lacticaseibacillus casei* has immunomodulatory activity and can enhance the intestinal mucosa. The ability of *Lacticaseibacillus casei* to produce EPSs is strongly associated with glycosyltransferase [29,30], which can be used in the production of EPSs from fermented milk. EPS synthesis comprises three essential steps—the uptake of a carbon substrate, intracellular synthesis, and excretion outside the cell [3,16,17]. The C/N ratio plays a significant role in EPS synthesis and varies depending on the strain used. For instance, [31] used lactose and mannose in a 7:1 carbon-to-nitrogen ratio to obtain a homopolysaccharide composed of galactose residues from *Lactobacillus plantarum*. In 2015, [14] produced an EPS heteropolysaccharide using a C/N ratio of 5. This EPS comprised glucose and mannose residues and exhibited both antitumor and antioxidant properties. In a similar study, [32] used *Lacticaseibacillus casei* to create two EPS samples. The first sample contained galactose, glucose, and arabinose, whereas the second contained galactose, glucose, and mannose. Once the EPS molecule was extracted and purified, it was submitted to different tests for characterization. One such test was gas chromatography–mass spectrometry (GCMS), which determined the mass of the polymer. Nuclear magnetic resonance (NMR) was also used, which identified hydrogens and carbons in the EPS structure, helping to identify the structural composition of the molecule (that is, whether it was monomeric or polymeric) [13,27,33,34]. EPSs can be used as natural, harmless prebiotic food additives in different sectors, such as the food, pharmaceutical, and environmental industries. Studies conducted by [2,12,35,36,37] demonstrated that EPSs have electrostatic affinities, which makes them effective adsorption agents for heavy metals in decontamination processes. EPSs can also serve as a potential reducer of trace elements [28,38]. Ref. [28] and colleagues (2020) conducted a study demonstrating that EPS produced by *Bacillus* spp. can adsorb Cd (II), Cr (VI), and Cu (II). Similarly, [39] and colleagues (2020) found that EPS produced by *Halobacillus* spp. has active hydroxyl, amino, and carbonyl groups that form complexes with heavy metals. These complexes protect the bacterial cell from the toxicity of the metal ions, ensuring resistance to these compounds. Ref. [40] and colleagues analyzed the composition of an EPS and found that it was composed of heteropolysaccharides, such as mannose, glucose, xylose, and rhamnose. The authors also found that the EPS had emulsification and flocculation activities and was resistant to multiple heavy metals. This suggests that this polymer could be used for biotechnological applications, such as the removal of metals. According to [41], new research is focused on using polysaccharides as reducing agents and stabilizers for metal nanoparticles. In the food industry, these polysaccharides have attracted interest as natural food additives. Due to their rheological properties, these substances can affect viscosity as well as the stabilizing, emulsifying, or gelling properties of fermented products, such as yogurt and cheese, thus enhancing texture and mouthfeel [13,21,42]. Polysaccharides also have favorable thermal properties, making these substances useful during thermal processes [42]. EPS obtained from *Lactobacillus* has probiotic properties, and a study by [2] showed an innovative approach to using it in the production of nutraceuticals. Strains of *Lactobacillus* have been shown to influence the human gastrointestinal microbial ecosystem and counteract physiological and immunological changes associated with diseases [43,44]. Several health benefits have been associated with the consumption of EPS produced by *Lacticaseibacillus casei*, such as anti-cancer, antioxidant, prebiotic, immunomodulatory, and cholesterol-lowering properties, as well as the prevention and relief of diabetes mellitus [14,27,45,46,47]. Therefore, the aim of the present study was to produce EPS using glycerol and molasses as carbon sources in the production process.

## 2. Materials and Methods

### 2.1. Screening for Potential EPS-Producing Bacteria

EPS production was studied for 22 strains. These strains were isolated in previous work and are stored in the Industrial Microbiology Laboratory (Appendix A). The 22 isolates, which were stored in 20% glycerol (Labsymth, Brazil) at −80 °C, were reactivated in De Man, Rogosa, and Sharpe (MRS Broth, BD^TM^ Difco^TM^, Franklin Lakes, NJ, USA) medium for 48 h and plated to determine the existence of pure colonies. Next, a 10% inoculum was again inoculated in an MRS medium. After 18 h of fermentation, the exponential phase of the microorganisms was obtained, and 10% of it was inoculated in a specific medium for EPS production. The inoculum was stirred at 130 rpm and 37 °C for 40 h.

### 2.2. Culture Media

#### 2.2.1. MRS Culture Medium

The medium for the microorganism pre-inoculum was composed of (in g.L^−1^) meat extract (10), yeast extract (5), dextrose (20), polysorbate 80 (1), ammonium citrate (2), sodium acetate (5), manganese sulfate (0.05), magnesium sulfate (0.1), and dipotassium phosphate (2).

#### 2.2.2. Semi-Defined Medium (SDM)

The SDM used to create the fermentation inoculum contained the following ingredients per liter: 10 g of bacto-casitone, 5 g of yeast nitrogen base, 2 g of ammonium citrate, 5 g of sodium acetate, 0.1 g of MgSO_4_·7H_2_O, 0.05 g of MnSO_4_, 2 g of K_2_HPO_4_, 10 mL of glycerol, 2 mL of glucose (20 g/L), and 1.0 mL of Tween−80. The pH was adjusted to 6.6 using 1 M NaOH or 1 M acetic acid solutions, modified from [13]. To produce EPS using molasses as the carbon source, the same substrates were used along with the same concentrations, but glycerol and glucose were replaced with 20 g·L^−1^ of molasses, following the method described by [13].

### 2.3. Fermentation Process

#### 2.3.1. Flask Experiments

This experiment was carried out in a 500 mL Erlenmeyer flask containing 100 mL of modified SDS medium. Growth of the Ke8 isolate in glycerol and EPS (g/L) production was monitored from 0 to 120 h.

#### 2.3.2. Reactor Experiments

The Lacticaseibacillus casei [48] strain was added to 3 L of modified SDM medium in a bioreactor (Zeta model, Bioreactor System for Cell Culture 2 with Bio-t Mini^®^ Control System, Rapperswil, Switzerland) and left to incubate for 128 h at 37 °C with agitation at 130 rpm. The pH was automatically controlled at 6.6 using injections of 5 M NaOH and 1 M acetic acid solutions, without O_2_ injection.

### 2.4. Analytical Methods

#### Sample Processing

Samples were withdrawn at certain intervals throughout EPS production. When glycerol and glucose were used as the carbon sources, 50 mL samples were withdrawn every eight hours between 0 and 128 h. When molasses was used, 50 mL samples were withdrawn every 12 h between 0 and 120 h. Optical density was measured by reading absorbance at 600 nm with the aid of a Bell spectrophotometer.

### 2.5. EPS Extraction and Assessment of Production

To extract crude EPS, 4% (*m*/*v*) trichloroacetic acid was added to 50 mL of fermented medium, and the mixture was stirred for 30 min at room temperature. Centrifugation was performed at 10,000× *g* and 4 °C for 15 min for cell removal and the precipitation of proteins. Double-volume cold ethanol was then added to the supernatant, and the mixture was stored at 4 °C for 24 h. The EPS produced was collected via centrifugation, dissolved in deionized water, and weighed on an analytical scale. The quantity of sugar in the EPS was estimated using the phenol–sulfuric acid method, using glucose as the standard. The quantity of EPS produced was determined by weighing the lyophilized EPS (Lyophilizer model L101, Liotop, São Carlos, Brazil). For the study involving molasses as the carbon source, EPS was extracted according to the method reported by [14], collected, and quantified in the same way.

### 2.6. EPS Isolation and Purification

The EPS was purified via dialysis against distilled water for 48 h at 4 °C using a dialysis membrane (Sigma 12 kDa, Merck KGaA, Darmstadt, Germany). The EPS was then lyophilized, fractionated through ion exchange chromatography using a DEAE-cellulose column (10 mm × 32 cm), and eluted with 0.2 M NaCl. The eluate was collected in 5 mL fractions, and the carbohydrate content was determined using the phenol–sulfuric acid method. Fractions containing polysaccharides were pooled, dialyzed, and lyophilized. To determine the total carbohydrate content, glucose was used as a standard, and the phenol–sulfuric acid method was employed. Total protein content was measured [49].

### 2.7. EPS Characterization—High-Performance Liquid Chromatography

To analyze the composition of a purified EPS sample, 100 mg of a given sample was hydrolyzed using 250 µL of H_2_SO_4_ (4%) at 121 °C for 30 min in an autoclave. The mobile phase (H_2_SO_4_ 0.005 N) was then added until obtaining 300 µL of the sample. The composition of the sample was analyzed using high-performance liquid chromatography (HPLC) (Shimadzu, Kyoto, Japan) with an IR detector and H_2_SO_4_ (0.005 N) as the mobile phase at a flow rate of 0.6 mL.min^−1^ at 65 °C. The HPLC column was Rezex ROA—Organic Acid H^+^ (8%).

### 2.8. Size Exclusion Chromatography

The molar mass distribution of EPSs was analyzed using an OMNISEC size exclusion chromatograph produced by Malvern Instruments. Viscotek RImax refractive index detectors from the same manufacturer were used for detection.

The samples were dispersed in a 0.1 mol.L^−1^ NaNO_3_ solution and filtered through 0.45 µm PTFE membranes. Three Shodex chromatographic columns (model SB-806 HQ) from SHOWA Denko (Tokyo, Japan) were used. The temperature of the columns was maintained at 40 °C throughout the analysis. Mobile phase injection consisting of 0.1 mol.L^−1^ of NaNO_3_ was used at a flow rate of 0.7 mL.min^−1^. To calculate the polymer mass, a calibration curve was obtained using polyethylene glycol (PEG) standards from Varian (Palo Alto, CA, USA). Molecular masses of these standards ranged from 224,000 to 195 g.mol^−1^, and the retention times were already known

### 2.9. EPS Analysis—Gas Chromatography–Mass Spectrometry

For analysis using gas chromatography–mass spectrometry (GC-MS), the EPS sample mixture was derivatized. First, an aliquot of 1 mg was hydrolyzed at 100 °C for 12 h in an oven (Fanen Retilinea, FANEN, Guarulhos, São Paulo, Brazil) with 200 to 500 µL of 1 M trifluoroacetic acid (TFA). After hydrolysis, the acid was removed via evaporation using a gentle blast of air. The hydrolysis products were suspended in distilled H_2_O (100 µL) and reduced with NaBH_4_ (pH 9.0–10.0) for approximately 12 h. Next, cationic resin (strongly acidic cation exchange resin (H^+^)—DIELAB) was added to the reaction, and the product at this stage was then filtered through cotton. The boric acid formed during the reaction was eliminated via co-evaporation with methanol in the form of trimethyl borate. The alditols obtained after removing the boric acid were acetylated with acetic anhydride and pyridine (1:1, *v*/*v*) at room temperature for 12 h. Alditol acetates were extracted by adding approximately 1 mL of chloroform (CHCl_3_), and the residual pyridine in the chloroform fraction was removed via complexation with 5% (*w*/*v*) copper sulfate. After complete removal of pyridine, the sample was filtered through cotton.

### 2.10. EPS Analysis—Fourier Transform Infrared Spectroscopy

The purified EPS was analyzed using Fourier transform infrared spectroscopy (FT-IR) to identify the largest structural groups. EPS spectra were obtained via the attenuated total reflectance (ATR) technique, using a zinc selenide crystal. The FT-IR spectra were recorded using a Perkin Elmer Frontier spectrometer, with 64 scans performed at a resolution of 4 cm^−1^ in the region of 4000 to 550 cm^−1^.

### 2.11. One-Dimensional 1 H Nuclear Magnetic Resonance (NMR) Spectroscopy

A Bruker Avance III 600 MHz spectrometer was used to measure the one-dimensional (1-D) NMR spectra of the EPS at room temperature. To perform the analysis, 10 mg of the purified EPS sample was dissolved in 630 µL of 99.96% D_2_O. Chemical shifts were expressed in parts per million (ppm).

The samples were identified through mass spectrometry using electronic impact ionization (70 eV). The Restek Rtx-5ms column (30 m × 0.250 mm × 0.25 μm) fused with silica was used. The interface and ion source temperature were maintained at 280 °C.

## 3. Results and Discussion

### 3.1. Fermentation of Ke8 Isolate (Lacticaseibacillus casei) in “Batch” Using Erlenmeyer Flask

Preliminary fermentation was carried out in a 500 mL Erlenmeyer flask before conducting studies at a larger scale of production in a bioreactor. The flask had a working volume of 100 mL and was placed on a bench shaking table. Growth of the Ke8 isolate in glycerol and the production of EPSs (g/L) were monitored from 0 to 120 h. These results are displayed in Figure 1. The first 12 h of incubation served as the adaptation period for the microorganism in the MRS medium with glycerol as a carbon source. After 12 h of fermentation, the Ke8 isolate had the highest growth rate (μmax = 0.52 h^−1^). These characteristics are consistent with the findings described by [13]. After 36 h of incubation, Ke8 entered its stationary and declining phase.

The Ke8 isolate had a maximum OD of 0.7 (Figure 1). In contrast, when [50] used glucose as a carbon source, the authors obtained a significantly higher maximum OD at 600 nm of 3. Ref. [16] reported that the use of different carbon sources can alter the composition and production of EPS due to the different metabolic pathways involved in carbon source metabolism. Glucose is considered to be the easiest carbon source to metabolize, as it is degraded into pyruvate and oxidized to form acetyl-CoA. However, some microorganisms may experience growth difficulties or disadvantages due to the limited absorption of glycerol into their cells [51]. EPS production was highest after 24 h of fermentation, producing approximately 4 g·L^−1^. The production profile indicates that the E PS is a primary metabolite produced during the exponential phase of microorganism growth, as described by [17]. This production profile was also observed by [50]. However, a decline in EPS production occurred between 24 and 42 h of fermentation, followed by a new peak at 48 h, which may be related to the nutritional conditions of the environment. EPS production is related to host stress, which can be caused by changes in the concentrations of carbon and nitrogen sources, pH, temperature, and other factors [4,5,17,52].

After successfully producing EPS using glycerol as a carbon source, the next step involved using Ke8 for EPS production with molasses as the carbon source.

### 3.2. Growth and Production Kinetics

Figure 2 illustrates the correlation between bacterial growth and EPS production during the fermentation process. The production of EPS was measurable from the onset of the fermentation process (0 h), with the highest amount, 1.8 mg L^−1^, recorded after 16 h of fermentation. However, a decrease in EPS production occurred in the exponential growth period of *Lacticaseibacillus casei* (16 to 32 h). This could be due to the consumption of EPS as a carbon source since it is composed of carbohydrates and proteins. Additionally, enzymes such as dextranase and glycosyl hydrolases were present, which degrade EPS, as reported in previous studies [17,48,53,54]. During the stationary phase and beginning of a second exponential phase of the microorganism, EPS production remained relatively constant. According to several studies, EPS can be produced when cells are growing [17,55,56] and can occur simultaneously in the beginning and middle of the exponential phase. EPS can also be produced at different stages during the fermentative process either as a primary or secondary metabolite [17]. After 56 h of fermentation, the microorganism began growing again, whereas the production of EPS decreased. This decrease may be related to the use of EPS as a carbon source [17]. After 104 h of fermentation, the EPS produced was characterized as a secondary metabolite. This finding was also reported by [13], who showed that EPS production occurred during the period when the microorganism had reached the stationary phase.

Preferably, a strain should produce EPS within 16 h of fermentation. However, EPS production decreased rapidly as the *Lactobacillus casei* Ke8 cells grew, which limits their practical applications. Studies have shown that a decline in EPS yield is due to enzymatic degradation or changes in the physical and/or chemical characteristics of the culture medium. A reduction is generally found in the stationary phase or the declining phase of growth. Ref. [57] and colleagues obtained 500 mg. L^−1^ of EPS from *Lactobacillus rhamnosus* using glucose or lactose as carbon source. The authors noted that EPS production was partially linked to the growth of the microorganism, with a decline in production as the fermentation process continued. The authors also discovered a depolymerizing enzyme in the fermentation medium by analyzing the degradation of EPS during fermentation through its molar mass and viscosity. The researchers found a wide range of glycohydrolases in their study on enzyme production profiles (α-D-glucosidase, β-D-glucosidase, α-D-galactosidase, β-D-galactosidase, β-D-glucuronidase, and traces of α-L-rhamnosidase). Two of these enzymes were found in this study: α-D-glucosidase and β-D-glucuronidase. In 1992, [58] conducted a study on different strains of *Lactococcus lactis* spp. and *Lacticaseibacillus casei*, which were capable of producing EPS in quantities ranging from 30 to 600 mg L^−1^. Ref. [59] reported that *Streptococcus salivarius* spp. could also produce EPS, and optimal production occurred when lactose or sucrose was used as the carbon source under the defined temperature and initial pH conditions. The researchers also discovered that environmental factors could impact the production and degradation of EPS by microorganisms and identified degradative systems of polysaccharides.

Although it is challenging to explain the decrease in EPS content and biomass observed at an early stage in this study, a high growth rate (OD [600 nm]: 1.5) can have a negative impact on EPS stability and biosynthesis, as a bacterium can quickly break down EPS to fulfill its nutritional needs for growth. Conversely, the biosynthesis of EPS is stimulated by a reduction in growth rate [58,59,60]. Different bacterial strains produce varying amounts of EPS. Lactic acid bacteria (LAB) generally have the lowest yields, ranging from 0.01 to 2.0 g L^−1^ [30,35]. Lactobacillus produces the least EPS, generating less than 1.0 g. L^−1^ [35]. This occurs because LAB acidify the culture media to a pH below 5.0, activating glycosyl hydrolase, which results in the enzymatic digestion/hydrolysis of EPS, significantly reducing the final yield [35]. Despite producing less EPS, the use of LAB is becoming increasingly popular due to its classification as being “generally recognized as safe”, making it a crucial natural alternative. Therefore, EPS-producing strains have enormous commercial potential [47].

In the present study, Erlenmeyer flasks provided better conditions for producing EPS from molasses using the *Lacticaseibacillus casei* strains (Figure 3). The data displayed in Figure 1 show the relationship between EPS concentration and bacterial cell growth as a function of fermentation time. The production of EPS was linked to bacterial growth and continued to increase for 48 h, which was when maximum production of the biopolymer was achieved. After this period, however, a decrease in production occurred. During the fermentation process, bacterial growth and biopolymer production exhibited a similar trend. Optical density (OD) reached 1.4 after 48 h of fermentation, indicating maximum growth and biopolymer production of the *Lacticaseibacillus casei* strain. The adaptation phase lasted for up to 12 h of fermentation. The EPS produced is considered a growth-associated metabolite. Using molasses as the sole carbon source proved to be promising for cell growth and EPS production. After 48 h of fermentation, approximately 140 mg L^−1^ of EPS was obtained, with an OD (600 nm) of approximately 1.4.

### 3.3. Extraction and Purification of EPS

A 100 mL sample was collected from the DEAE-cellulose column; this volume is equivalent to 10 times the total column volume. This sample was dialyzed, lyophilized, weighed, and used for the physicochemical characterization. To calculate the carbohydrate and total protein content, a straight-line equation was used, with bovine serum albumin serving as the standard for quantifying proteins and using glucose for carbohydrates. Samples with the highest absorbance were combined to calculate the carbohydrate concentration. The carbohydrate content was found to be 39.4% when glucose and glycerol were used as a carbon source and 52.9% when molasses was used. Similarly, the protein content was 18% when glucose and glycerol were used and 11.69% when molasses was used for EPS production.

In a semi-defined medium, [13] reported a total EPS production amounting to 0.09 g·L^−1^, with 92.35% carbohydrates and 1.38% proteins, using glucose as the carbon source. Ref. [61] reported a higher EPS yield of 1.67 g·L^−1^. Ref. [36] obtained 18 mg·mL^−1^ of EPS composed of 22% carbohydrates and 2% proteins in a medium in which soy milk was used as the carbon source. It is important to note that the EPS production in this study, using glycerol and glucose as a carbon source, was efficient, yielding 3.0 g·L^−1^ (m/m). With the use of molasses, the yield was 3.5 g·L^−1^ (m/m), which is substantially higher than the yield reported in previous studies.

### 3.4. Analysis of EPS Produced from Glycerol + Glucose and Molasses Using HPLC

The HPLC analysis of the EPS obtained from *Lactobacillus casei* using glycerol and glucose as carbon sources revealed a heteropolysaccharide consisting of glucose and mannose. The retention times of glucose and mannose were 10.416 min and 10.8 min, respectively (Figure 4). The characteristic peaks had areas of 18,119 and 14,668, heights of 923 and 880, and concentrations of 0.066 g·L^−1^ and 0.046 g·L^−1^, respectively. This type of EPS has also been discovered by researchers using other microorganisms and skimmed milk, lactose, galactose, glucose, and sucrose as carbon sources [14].

The EPS obtained from *Lacticaseibacillus casei* when molasses was used as the carbon source was classified as a heteropolysaccharide composed of glucose, arabinose, and galactose, with retention times of 9.767 min, 11.679 min, and 11.086 min, respectively (Figure 5). The characteristic peaks had areas of 303,076, 1,173,146, and 3,465,224, with heights of 11,857, 43,237, and 144,188, respectively. The glucose, arabinose, and galactose concentrations were 2.245 g L^−1^, 4.742 g L^−1^, and 14.290 g L^−1^, respectively. In a recent study involving the growth of *Lacticaseibacillus casei* in an SDM medium with molasses as the carbon source, ref. [62] discovered an EPS consisting of carbohydrates such as galactose, arabinose, glucose, and mannose. This EPS mainly comprised the monosaccharides galactose and glucose [62]. In a study by [58], *Lactobacillus casei* spp. produced heteropolysaccharides when cultivated in a medium containing milk and subjected to ultrafiltration. The EPS primarily consisted of galactose and glucose, with small quantities of mannose, rhamnose, and pentoses.

### 3.5. Exclusion Chromagography

The size exclusion chromatograms for the EPS produced from glycerol (Figure 6A) and molasses (Figure 6B) are displayed in Figure 6. The refractive index detector is represented by the black curve in both plots. The red curve corresponds to the PEO standard, whose molecular weight is known. A shoulder with a retention volume of 30.87 mL and a maximum retention volume of 32.56 mL is shown in the chromatographic profile in Figure 6A, which indicates that separation was not complete. Glycosidic units (180 g mol^−1^) that were not polymerized correspond to the greatest retention volume, which has a molar mass similar to that of the lowest injected standard. In terms of the retention volume of 30.87 mL, the molar mass equals 1730 g mol^−1^. The chromatographic profile displayed in Figure 6B has a shoulder at 31.50 mL, and the EPS obtained from molasses had a molar mass of 1182 g mol^−1^.

### 3.6. Composition of Monosaccharides in EPS Based on GC-MS Analysis

To determine the structural composition of the molecule, the EPS chromatograms obtained using GC-MS were compared to chromatograms of sugar standards, such as mannose, rhamnose, arabinose, fructose, glucose, and galactose. The library of the National Institute of Standards and Technology (NIST) was also used for comparison. Due to the low concentration of carbohydrates in the EPS molecule, it was necessary to monitor ions in standard sugar samples to detect them in the EPS sample. Thus, the following precursor fragment ions were monitored: 115, 128, 139, 145, 170, and 187 *m*/*z*.

Figure 7 shows the composition of the EPS obtained from *L. casei* using glycerol and glucose as carbon sources. The sample contains ions 115, 128, 139, 145, 170, and 187 *m*/*z*. The mass spectrum obtained through ion monitoring was compared to the NIST library contained in the equipment itself. The GC-MS analysis of the composition of the EPS from *L. casei* suggested that it consists of D-mannitol and D-glucitol (mannose and glucose). EPSs from lactic-acid-producing bacteria can be used in the food, pharmaceuticals, and cosmetic industries and also have environmental applications. These natural, harmless additives can serve as adsorption agents for heavy metals and can even reduce trace element concentrations. Studies report that EPS produced by *L. plantarum* can inhibit the activity of α-amylase in the pancreas, giving it potential for application in preventing and relieving diabetes mellitus. EPSs composed of mannose, rhamnose, glucose, and xylose have emulsification and flocculation activities and can be applied in the biotechnology industry to remove metals. Other studies have shown that EPSs composed of mannose and glucose can form complexes with heavy metals and protect the bacterial cell from zinc, cadmium, lead, nickel, and copper toxicity. EPSs composed of galactose, glucose, and mannose have also been found to improve the freezing and thawing survival ability of microbial isolates and exhibit anti-tumor effects against colon cancer cells. These EPSs have potential applications in food products and the medical field.

### 3.7. Analysis of EPS Using FT-IR

Figure 8 presents the FTIR spectra of exopolysaccharides produced by *L. casei* using molasses (Figure 8A) and glucose + glycerol (Figure 8B). Both spectra showed similar signals. The elongated, broad, intense bands in the region of 3400 cm^−1^ indicate the presence of the hydroxyl group (^-^OH) of carbohydrates. The absorption peaks at 2926 and 2935 cm^−1^ (Figure 8A,B, respectively) were attributed to the asymmetrical C-H stretching vibration of the methyl group. The absorption peaks at around 1640 cm^−1^ indicate the characteristic IR absorption of polysaccharides [63] and correspond to the stretching vibration of the carbonyl group (C=O) (Singh et al., 2011). The samples exhibited absorbance in the region of 1200 to 970 cm^−1^, which is considered the region in which exopolysaccharides are identified [64]. The bands around 1000–1200 cm^−1^ were attributed to C–O–C stretching characteristic of α-pyranose carbohydrates [65] and indicate the presence of α-1–6 glycosidic bonds [14,66].

### 3.8. Analysis of EPS Using NMR

Figure 9 displays the ^1^H NMR spectra of exopolysaccharides produced by glucose + glycerol (Figure 9A) and molasses (Figure 9B).

According to the literature, the ^1^H NMR spectrum of a polysaccharide can generally be divided into three major regions: the anomeric region (δ H 4.5–5.5 ppm); the ring proton region (δ H 3.1–4.5 ppm), which is assigned to protons attached to C_2_-C_6_; and the alkyl region (δ H 1.2–2.3) [14,31]. The ^1^H NMR spectra of the EPS produced from glucose + glycerol (Figure 9A) and molasses (Figure 9B) showed some signals in similar regions. In both spectra, signals were found in the region between 3.0 and 4.0 ppm, a region corresponding to protons linked to C_2_ and C_6_ of the glycosidic structure. Signals were also found between 1.0 and 2.0 ppm, which are characteristic of alkyl groups. A difference between the spectra was found in the anomeric carbon region. For EPS produced with glucose and glycerol, no signals were found in the anomeric region due to the presence of water, which suppressed the signals. For the EPS produced with molasses, a doublet (J = 3.00 Hz) of anomeric hydrogen was found at 4.99 ppm. According to the literature, the coupling constant observed is characteristic of the alpha conformation (α) from glucopyranose [14,67].

## 4. Conclusions

The procedure performed in the present study proved to be efficient, and the results revealed the behavior of the microorganisms in the production medium in terms of the production of EPS, pH, and sugars present in the samples. All 22 isolates tested produced EPS. Moreover, the microorganism *Lacticaseibacillus casei* was able to produce EPS using glycerol + glucose and molasses as carbon sources. The EPS obtained when using glucose and glycerol as carbon sources was characterized as a heteropolysaccharide composed of glucose and mannose. When molasses was used as the carbon source, the composition of the EPS was glucose, galactose, and arabinose. The molar masses of the EPS were 1730 g.mol^−1^ when glucose and glycerol were used and 1182 g.mol^−1^ when molasses was used. NMR and FT-IR analyses confirmed the existence of polysaccharide characteristics in the structure of the EPS produced.

## Figures and Tables

**Figure 1 microorganisms-12-01159-f001:**
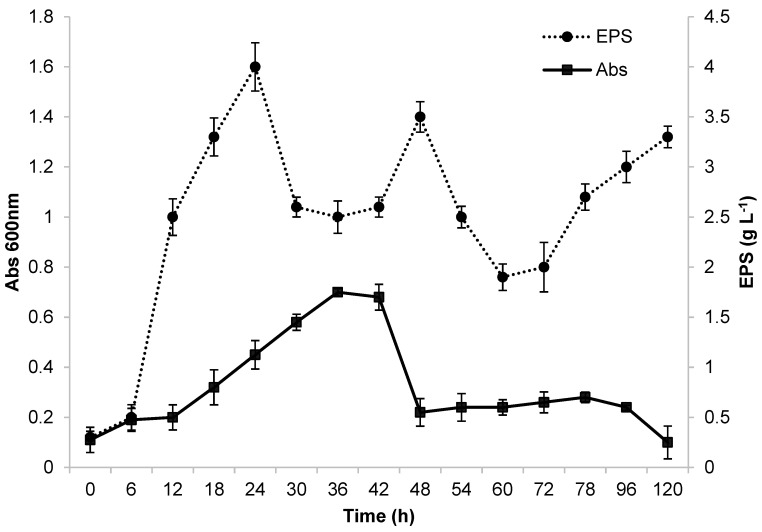
Growth and EPS production by *Lactticaseibacillus casei* Ke8 in Erlenmeyer flask at 130 rpm, 37 °C, and pH 6.6.

**Figure 2 microorganisms-12-01159-f002:**
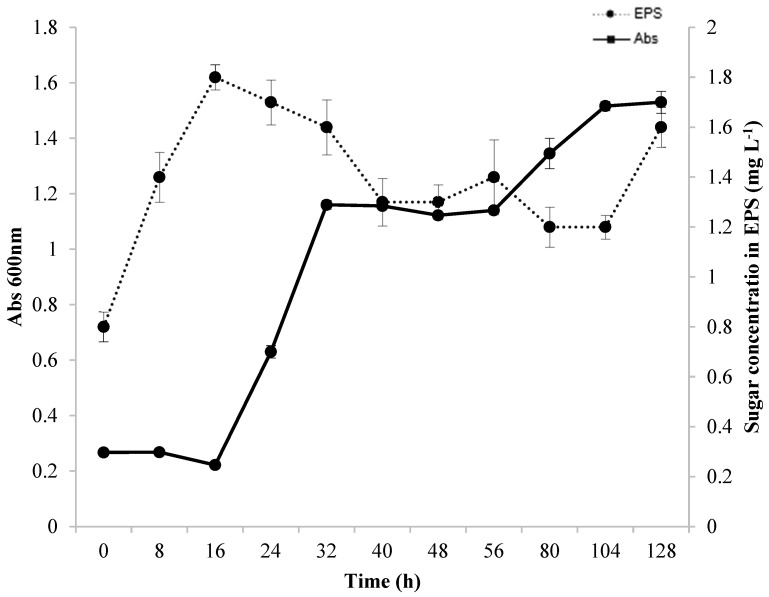
Bacterial growth and EPS production kinetics in SDM medium during 128 h of fermentation at 37 °C under agitation at 130 rpm in a medium containing glycerol and glucose as carbon sources.

**Figure 3 microorganisms-12-01159-f003:**
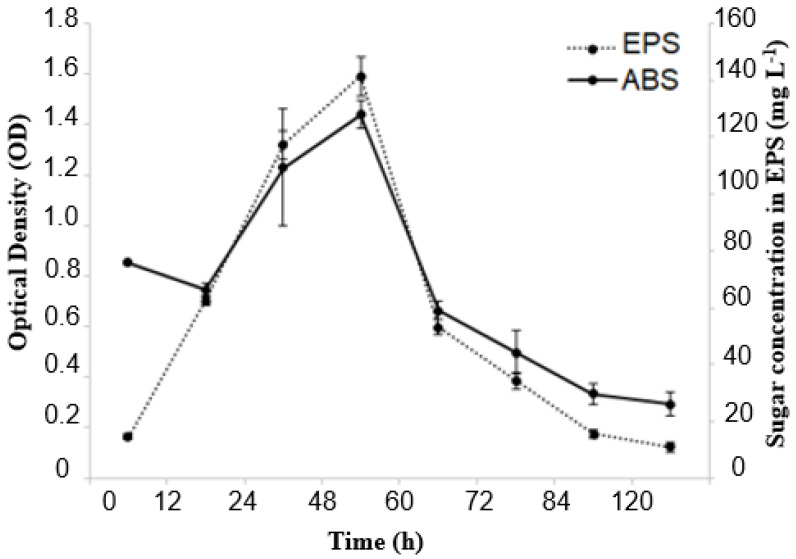
Bacterial growth and EPS production kinetics during 120 h of fermentation at 37 °C under agitation at 130 rpm in a medium containing molasses as a carbon source.

**Figure 4 microorganisms-12-01159-f004:**
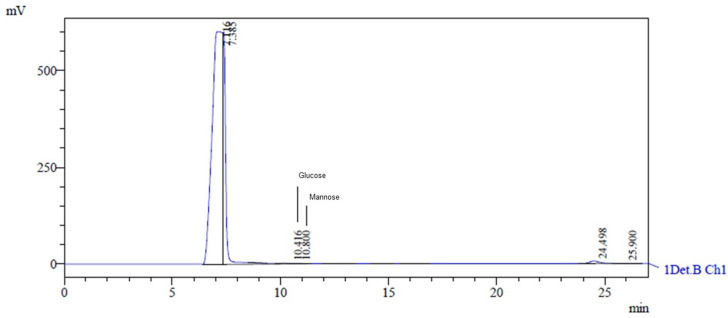
HPLC chromatogram showing monosaccharide composition of hydrolyzed EPS. The peak at 10.416 corresponds to glucose, and the peak at 10.800 corresponds to mannose.

**Figure 5 microorganisms-12-01159-f005:**
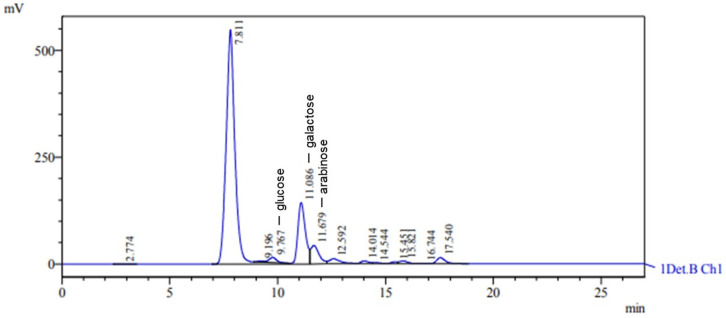
HPLC chromatogram showing monosaccharide composition of hydrolyzed EPS. Peaks at 9.767, 11.086, and 11.679 correspond to glucose, galactose, and arabinose, respectively.

**Figure 6 microorganisms-12-01159-f006:**
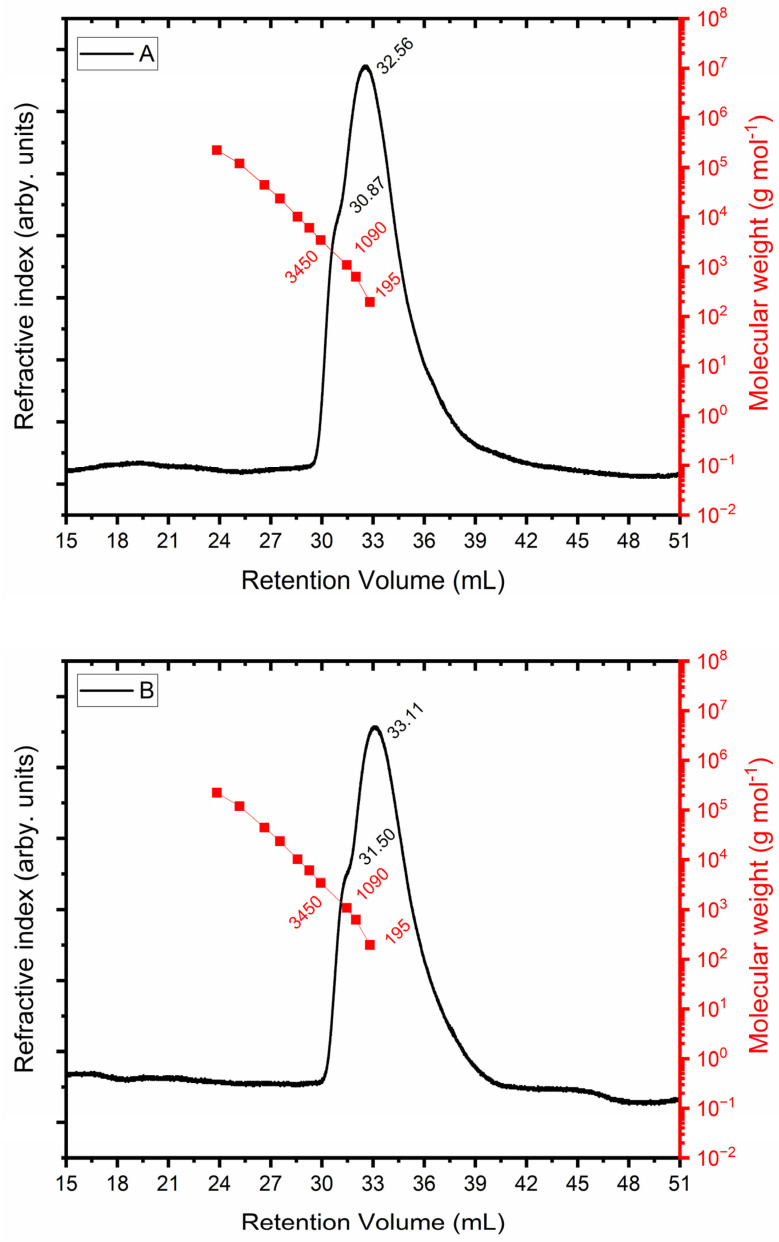
Determination of molar mass of exopolysaccharide using glycerol + glucose (**A**) and molasses (**B**) as carbon sources.

**Figure 7 microorganisms-12-01159-f007:**
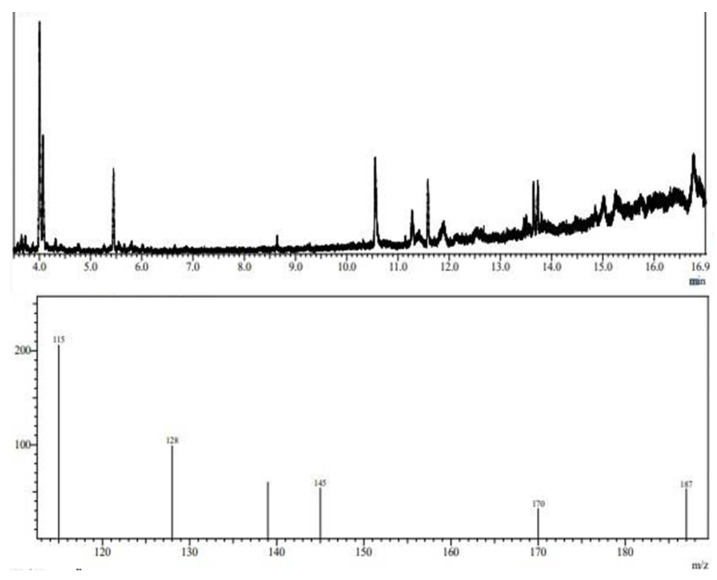
Fragmentation profile of alditol acetates after chemical derivatization of EPS molecule obtained from *L. casei*.

**Figure 8 microorganisms-12-01159-f008:**
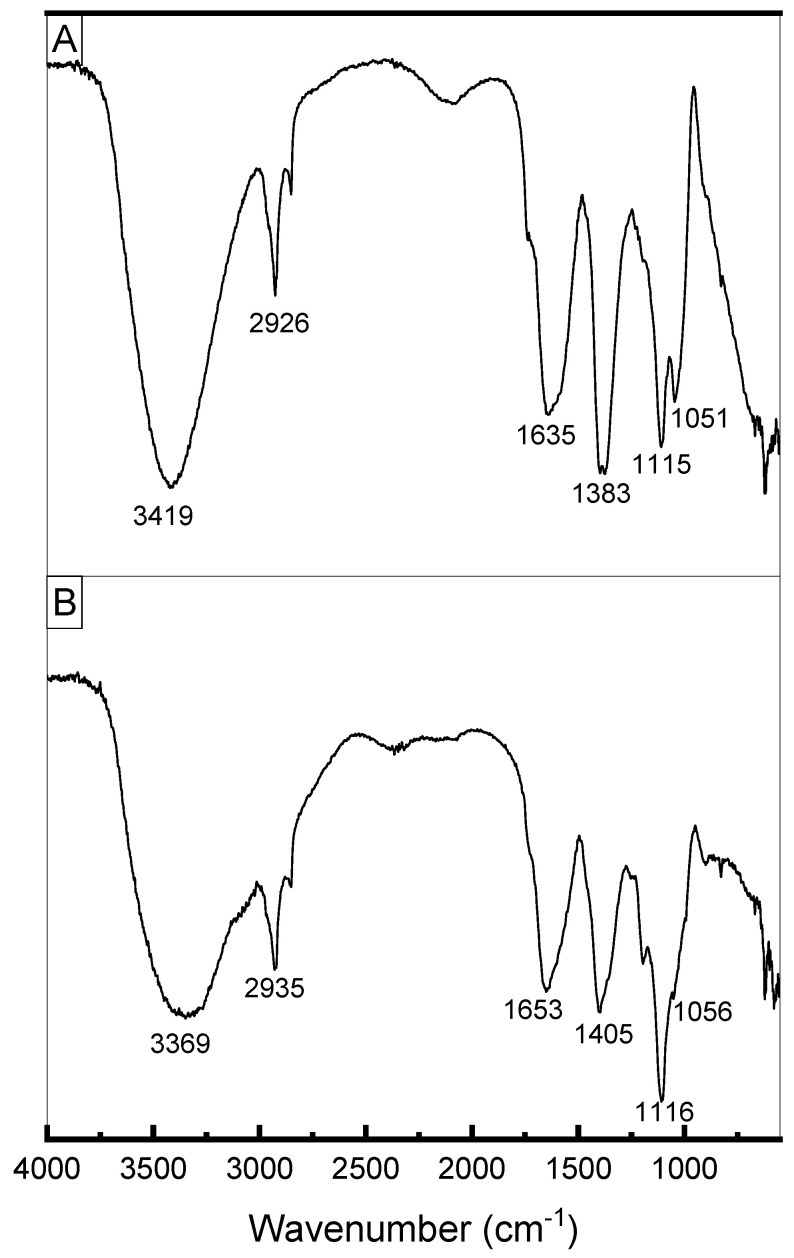
(**A**) FTIR spectrum of EPS produced by *Lacticaseibacillus casei* using molasses as carbon source. (**B**) FTIR spectrum of EPS produced by *Lacticaseibacillus casei* using glycerol + glucose as carbon sources.

**Figure 9 microorganisms-12-01159-f009:**
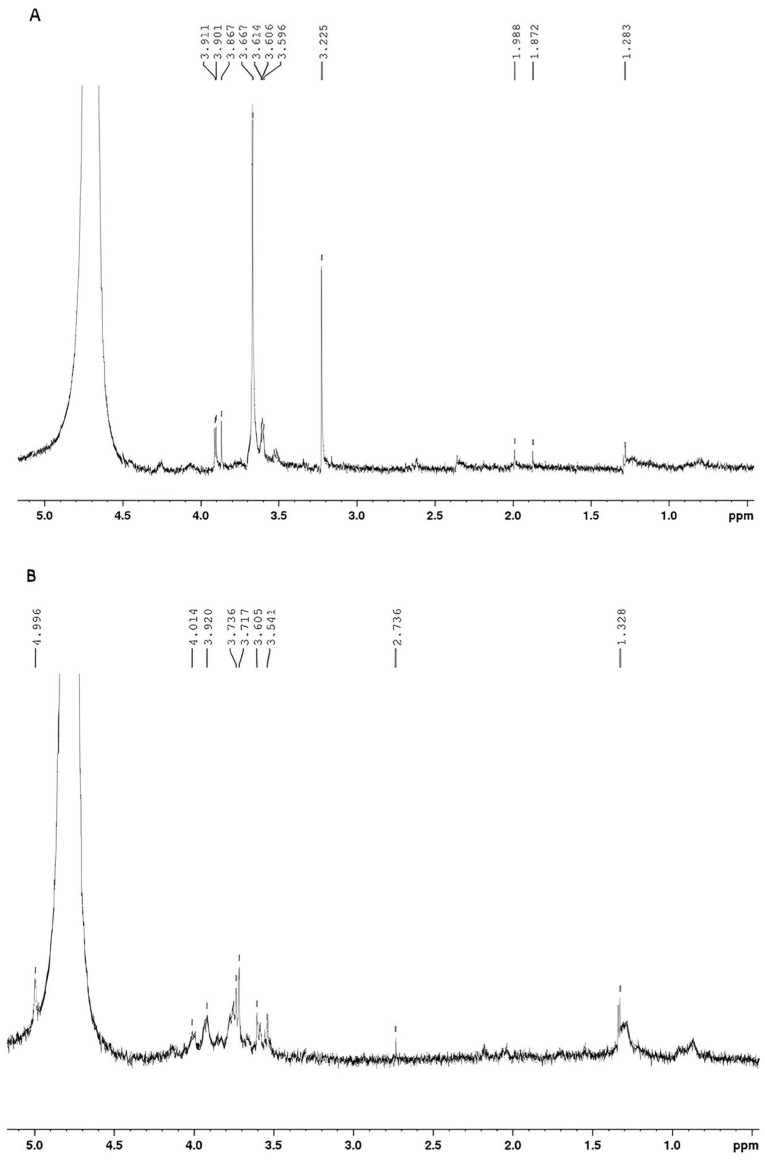
(**A**) ^1^H NMR spectrum of EPS produced by *Lacticaseibacillus casei* using glycerol + glucose as carbon sources. (**B**) ^1^H NMR spectrum of EPS produced by *Lacticaseibacillus casei* using molasses as a carbon source.

## Data Availability

Data are contained within the article.

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
