# Peer review of "EPS Production by Lacticaseibacillus casei Using Glycerol, Glucose, and Molasses as Carbon Sources"

_microorganisms, 2024, doi:10.3390/microorganisms12061159_

Round 1
Reviewer 1 Report
Comments and Suggestions for Authors
Recently, many studies have focused on cheap alternative carbon sources, diverse strains of producing microorganisms, as well as on improving their culture conditions for the purpose of effective EPS production. This problem is important and relevant. However, the manuscript needs to be improved.
Some comments are:
1. Pg 1, line 11-13: «Screening was carried out with 22 strains of Lactobacillus to determine that which achieved the highest production of EPS based on dry weight at 37 °C».
Why is this sentence in the Abstract if the results on screening and yield of EPS with 22 strains are not included in the article?
2. Pg 1, line 13-16: «Fermentation was performed in triplicate in a bioreactor using three liters of modified semi-defined medium. EPS was extracted, dialyzed, lyophilized, and purified. The total quantity of EPS was measured based on the concentration of total sugars in the samples using the phenol-sulfuric acid method. Protein content was quantified according to the Lowry method».
In my opinion, there is no need to write in the Abstract about the number of repetitions and in detail about all the methods.
3. Specify correct units of measurement throughout the text: «g.mol-1, g.L-1, g/L, mL.min-1, mol. L-1».
4. Pg 1, line 21-22: «The molecule was characterized using HPLC, size exclusion chromatography, GC-MS, FT-IR, and 1H NMR».
It is advisable to decipher the abbreviation of methods.
5. The aim of the study must be written in the Abstract.
6. Pg 1, line 23-24: «The results suggest that Lactobacillus casei can grow in media using alternative carbon sources and can produce EPS».
The conclusion in the Abstract should be changed to something more meaningful, emphasizing the novelty of the research.
7. The word exopolysaccharides should be added to the keywords.
8. Chapter 1 Introduction is not well written. You need to rewrite.
9. Pg 2, line 45-47: «Bacterial cellulose is widely used as an emulsion stabilizer in the cosmetic industry, as artificial skin for medical applications, or as an acoustic membrane [55, 4, 26]».
It is not clear why the authors give the example of bacterial cellulose when the article is devoted to the EPS of lactic acid bacteria. In addition, the literature references do not correspond to the text: 4. Badel, S.; Bernardi, T.; Michaud, P. New perspectives for Lactobacilli exopolysaccharides. Biotechnol. Adv., 2011, 54–66. 26. Hu, X.; Pang, X.; Wang, P. G.; Chen, M. Isolation and characterization of an antioxidant exopolysaccharide produced by Bacillus sp. S-1 from Sichuan Pickles. Carbohydr. Polym., 2019, 9–16. 55. Vu, B.; Chen, M.; Crawford, R. J.; Ivanova, E. O. Bacterial extracellular polysaccharides involved in biofilm formation. 626 Molecules, 2009, 2535–2554.
10. Pg 2, line 55-59: «EPS is made up of various biochemical groups that are secreted by different microorganisms, including prokaryotes and eukaryotes. The polymer mainly consists of polysaccharides or carbohydrates, proteins, lipids, phosphates, acetates, succinates, nucleic acids, and humic acids [52]. Carbohydrates and proteins comprise around 90% and 75% of the EPS molecule, respectively [38]».
The article by Tiwari et al. (2020) [52], which the authors refer to, EPS means Extracellular polymeric substances (EPS). "Extracellular polymeric substances (EPS) are biopolymers, composed of polysaccharides, nucleic acids, proteins and lipids." The authors of this article define EPS as exopolysaccharides. Therefore the proposals are incorrect.
11. Pg 2, line 61-63: «Polysaccharides are carbohydrates that can be found in different forms, such as dextran, cellulose, or curdlan, which is a gum made up of (1,3)-β-d-glucan».
Dextran is an α-D-glucan, cellulose is a β-1,4-D-glucan.
12. Pg 2, line 63-65: «Heteropolysaccharides, such as alginate, xanthan gum, gellan gum, benzol gum, cholanic acid, and hyaluronic acid, are usually produced by lactic-acid bacteria».
Xanthan and a number of other EPSs are not produced by lactic acid bacteria.
13. Pg 2, line 93: I would like it to be written not “[62] and colleagues”, but Zeng and colleagues.” This occurs throughout the text.
14. I recommend adding information in the Introduction about nutrient media and carbon sources for EPS biosynthesis by Lactobacillus casei, and EPS production by other bacteria using glycerol, and molasses as carbon sources. After all, the aim of the present study was to produce EPS using glycerol and molasses as carbon sources in the production process.
15. Pg 4, line 141: «2 mL of glucose». Why not g/L?
16. Pg 4, line 163: «Ethanol was then added to the supernatant …». In what quantity?
17. Pg 5, line 164: «The EPS produced was collected by centrifugation, dissolved in deionized water, and weighed on an analytical scale». It is not clear what was weighed.
18. Why is Table 1 needed if the article only presents the results of studies with the Ke8 isolate (L. casei).
19. I noticed the authors did not use any statistical analysis. I would recommend it in order to improve the quality of the paper.
20. I recommend to change the title of chapters 3.1, 3.2 and 3.3. The content of these chapters needs to be written more clearly and logically.
21. Figure 1 is difficult to understand. The authors need to put more information in the figure legend. Standard deviations are not indicated in the figure.
22. Pg 6, line 226-227: «This could be due to the consumption of EPS as a carbon source, since it is composed of carbohydrates and proteins».
What do you mean by EPS? Exopolysaccharides do not consist of proteins.
23. It is necessary to mark the curves in Fig. 2. The signature on the right is unclear.
24. ABC must be deciphered in the caption to Figure 3. In general, captions to all figures should be more informative: what kind of medium, flasks or bioreactor, etc.
25. The quality of figures 4, 5, 7 and 9 needs to be improved.
26. Pg 16, line 470-472: «The procedure performed in the present study proved to be efficient and the results revealed the behavior of the microorganisms in the production medium in terms of the production of EPS, pH, and sugars present in the samples». There are no pH data in the article.
27. Pg 16, line 472-473: «All 22 isolates tested produced EPS». There is no data in the article about 22 isolates. Information is provided on only one, the Ke8 isolate (L. casei).
28. The conclusion needs to be rewritten. It should be changed to more meaningful, emphasizing the novelty of the research.
Author Response
Please see comments below.

Reviewer 2 Report
Comments and Suggestions for Authors
1. The introduction is very long, it is necessary to adjust it only with the most important things that impact the substantial results. Since his work focuses on the production of EPS, I suggest that he should talk about the metabolic mechanisms of synthesis of EPS by referring to the C/N ratio and include how they influence the metabolic flux by reference to the carbon and nitrogen content of Lactobacillus casei, or in his case the general composition of a microorganism. Clarify how nutrient deficiency influences metabolism (or correct ratio of carbon and nitrogen). Elucidating the effect of C/N could address the effect of the carbon source on EPS synthesis, this would help clarify fermentation processes for EPS production.
2. Line 125-127. You should give more details about the method used to determine the EPS production capacity (screening). It is not indicated what the system (tube, flask, etc.) is, nor what is the result that can be useful in determining the capacity of the strains.
3. Table 1. in the LAB isolation section, it should only describe the coordinates and the basics of sampling. How were the strains identified?
4. In the fermentation section, you should describe all the variables you used, for example what concentration of inoculum did I adjust the system to?, what was the C/N ratio?.
5. Section 2.4 indicates that a biochemical characterization was performed and only presents a catalase test (this is insufficient since there are complementary biochemical tests) and mixes a description of the test that is unnecessary; only the methodology should be written in such a way that any reader can reproduce it.
6. In section 2.5, what ratio of ethanol was used to precipitate EPS and under what conditions, cold, stirring, static?
7. Line 252. You indicate a specific rate of growth, however in materials and methods it is not described how the kinetic parameters were calculated. Gompertz, logistic?
8. Figure 1, Error bars are not shown in the graph, did they not perform the experiments in triplicate? What statistical method did you use to determine the significant differences between each point, or how did you define the maximums?
9. Figure 2. You must properly label the data on both the axes and the data. If the dotted line is biomass, as accounts for the decrease after 16 hours., you are using a method that does not distinguish between live dead cells or therefore there should be no decrease. What is your explanation?
10. If growth is correlated with production, what would be the contribution of this comparison?
11. Figure 3 is unclear, and does not indicate which DO was measured?
12. What are the statistical methods you used for your analyses?
13. Measured the consumption of the carbon sources used?, which were the controls used in the fermentation processes?
14. You indicate that using molasses and glycerol production is adequate, however there is no comparison with literature that can reinforce that assumption.
15. In its conclusion, it should clearly indicate what the contribution of the work was, avoid summarizing results.
Author Response
Please see comments below.

Reviewer 3 Report
Comments and Suggestions for Authors
The article is very unfinished, the authors should spend more time on analysing the data obtained, present it in the form of a table. Figures 4-7 are not informative. The results are not clear. It is necessary to discuss the change of EPS composition depending on the carbon source. The article is not up to the high standard of the journal Microorganisms.
Some remarks in the form of notes in a pdf file

Author Response
Revisor 3
Please see the attachment.

Round 2
Reviewer 1 Report
Comments and Suggestions for Authors
The authors responded to most of the reviewer's comments and made appropriate changes to the manuscript. But corrections to minor methodological errors and text editing are required.
Author Response
It could indicate where these corrections should be made. Attached is a declaration that the text was written in native English

Reviewer 2 Report
Comments and Suggestions for Authors
The document now can be accept
Author Response
Thank you
Reviewer 3 Report
Comments and Suggestions for Authors
The authors did not take into account the comments, the article was practically not improved. All previous comments remained.
Author Response
The comments were made in the PDF tab, please check them. If you have any questions, please point out the errors
line 229
Sorry for not informing you as I responded to the other reviewer who had asked the same thing. These results come from another work carried out with an Industry interested in making lactic acid polymers, and these microorganisms were isolated and carried out with the same tests using Biochemical Methodology and confirmed by Molecular Biology
this experiment is not described in the materials and methods section
We done
In Table 1 this strain had no species name, where did it come from?
We done
line 472
Where are these results?
These were preliminary data when testing to observe which microorganism produced EPS using glycerol as a carbon source.
We explained in the text that identification was through biochemical and molecular tests (16S rRNA)
